# Integrating Chatbot and Augmented Reality Technology into Biology Learning during COVID-19

**Chi-Hung Chuang** [1], **Jung-Hua Lo** [2,*] and **Yan-Kai Wu** [2]

1 Department of Information & Computer Engineering, Chung Yuan Christian University, Taoyuan City 320314, Taiwan
2 Department of Applied Informatics, Fo-Guang University, Yilan County 26247, Taiwan
* Correspondence: jhlo@mail.fgu.edu.tw

**Abstract:** The novel coronavirus (COVID-19) pandemic is rampant around the world, and teachers and students are unable to attend physical classes in the midst of a serious outbreak. This study aims to design a user-friendly, educational chatbot application interface that can be used as an after-school self-learning tool for students to enhance their interest and comprehension and increase the effectiveness of their learning at home. The system adopts the Chatfuel platform as the core interface and incorporates augmented reality technology to build a chatbot that allows users to interact with it after they have logged in to Facebook. The content is based on the biology subject of the first year of junior high school and is integrated into the online teaching with augmented reality teaching materials. A user survey is conducted to understand students' attitudes towards learning biology with the aid of the ARCS motivation model, with 102 valid questionnaires received. The results show that the AR-based chatbot system developed in this study significant influenced the indicators in the ARCS motivation model; therefore, the intention to use the system is presumed to result in a noticeable increase in student learning outcomes when using the system. Accordingly, this study proposes new online learning tools for students to use at home during the pandemic, and the system also provides references for the future development and modification of educational chatbots.

**Keywords:** augmented reality; chatbot; ARCS (Attention Relevance Confidence and Satisfaction) model; e-learning

## 1. Introduction

### 1.1. Research Background and Motivation

In the past three years, many industries around the world have been affected by the ravages of COVID-19. Even the education industry, which is considered to be relatively stable and less volatile, cannot escape the impact of the epidemic. In order to reduce the risk of cluster infection and maintain appropriate social distance and to ensure that education is not interrupted, many changes have been made in education around the world, such as the conversion of traditional classrooms to online distance teaching. According to UNESCO, the COVID-19 pandemic, which has affected more than 1.5 billion students, has hit the most vulnerable learners the hardest. Currently, most schools are back open worldwide, but education is still in recovery, assessing the damage done and lessons learned [1]. Furthermore, during this epidemic, students in Taiwan are learning courses through distance platforms, using technology tools, such as Zoom, Team viewer, Google Meet, Zuvio, etc. to conduct classroom teaching. However, when distance teaching is implemented, it is often found that students who are on the computer side of their homes are prone to poor interaction with teachers and thus feel limited in their learning. Fortunately, with the rapid development of information technology, the application of multimedia in learning has become more and more diverse. Through the use of mobile apps combined with augmented reality (abbreviated AR) technology, the students' classroom learning has

become more interesting and will provide students with a more practical understanding of course content. With the current popularity of mobile phones, AR can be said to be a low-threshold information device demand [2–4], and it is also a feasible teaching tool in the teaching field and can improve students' learning motivation and learning performance [5–7]. Moreover, in the current school curriculum, it is a trend for teachers to combine mobile devices and use digital teaching materials for teaching. These interactive mobile devices and diverse resources are integrated into learning activities, a more lively and interesting way of learning than traditional textbooks, and also bringing a new trend to education. For example, AR technology combines text, sound, pictures, and web pages to make learning materials more vivid and interesting, allowing users to learn easily wherever they are and to clearly explain difficult-to-express knowledge.

The urgent need for the digitalization of global education brought about by the epidemic has also prompted teachers in Taiwan to think about curriculum transformation. In the TWNIC Taiwan Internet Report [8], it is observed that, in terms of educational applications, the age range of those using online learning is 15–24 years old, and the percentage is higher. The outbreak of indigenous cases of novel coronavirus (COVID-19) in Taiwan in May 2021 led to the closure of schools in several counties and cities, just as many countries were similarly afflicted. The severity of the situation prevented teachers and students from attending physical classes in school, underscoring the importance of online learning using digital materials. Through the software for natural-sciences-based e-learning ranging from e-book teaching, digital whiteboards, online video courses, to AR or virtual reality (abbreviated VR) interactive courses, the researcher has identified the possibility of building social software chatbots into digital teaching materials. The applications of social software are also very diverse. Facebook, which has been working on chatbot technology for many years, can build a variety of chatbot application scenarios through its Messenger chat software, which have the advantage of being fast in development and can be launched immediately after writing the program. Considering that students may have smartphones with different operating systems, e.g., Android or iPhone, the chatbot supports cross-platform devices to increase the convenience of operation. The developer can view the usage data through the back end and the time spent by students watching the lessons and interacting with them, which serves as ground for research in the development of learning-based chatbots to aid online learning after school.

At present, most of the web-based teaching platforms use videos to narrate, so it is hoped that interactive elements can be added to the study; in addition, given that the users were about 12 to 15 years old, a user interface with buttons and pictures was adopted to increase the convenience of intuitive operation. In this paper, a chatbot was built on the Facebook platform, making it easier to apply for a user account. On first run, a guide information page was added and the data initially set by the students was stored in the core chatbot database for the integrity of subsequent data. When managing the authentication, the chatbot can be logged into with account information, which is more compatible and does not require individual manual input to create a student account. A learner-based digital chatbot was designed through the Chatfuel platform to develop a system that would increase the convenience of after-school learning and self-directed learning. While developing and integrating AR digital educational materials, the reason for introducing AR technology for the biology curriculum is to address the problem that students mostly read physical textbooks and have less opportunity to come into contact with physical organisms through natural ecology. With the aid of AR digital learning materials, students are not restricted by time and place and can easily engage in independent learning after class. By adding the chatbot, students will be able to learn more about biology in the course of the operation process and interact with the digital teaching material; by operating the animal models, they can observe them in detail and understand their definitions and lifestyles, which is conducive to exploring natural creatures and piquing students' interest.

### 1.2. Research Purpose

The aim of this study is to look into the creation of an AR chatbot to be integrated into digital teaching materials for the subject of the natural science and to explore the use of the chatbot by students. Its content covered the biotaxonomy of the biology subject in the first year of junior high school and was applied to the system application of this chatbot. Students can use the chatbot with AR digital materials to gain knowledge and understanding of biology. It is hoped that, through practical assistance to school teachers and students in digital learning, the basic skills and literacy of students in operating computers and smartphones and other related devices will be developed and that the use of AR digital materials will spark their interest in learning natural sciences, increase their understanding and confidence in learning, and promote effective linkage in the field of biological knowledge, thus achieving a win–win situation. By constructing a chatbot with e-learning materials, this study focuses on the use of smart devices by a class of first-year students of a junior high school in Yilan County to engage in the e-learning of biology at home after school and analyzes the research objectives and issues for future study or the improvement of the chatbot e-learning materials. Thus, this study was conducted to respond to the following two questions:

1. How do we develop an effective online learning software so that students can study biology courses at home during COVID-19?
2. After students use this AR chatbot, how do we verify students' learning interest and effectiveness?

The rest of this paper is organized as follows: Section 2 describes the literature review of the study. The research model and hypotheses are proposed in Section 3. Section 4 discusses the research methodology. Section 5 discusses the results obtained. Finally, Section 6 contains conclusions and suggestions.

## 2. Literature Review

### 2.1. AR-Based Learning

Milgram et al. [9] regarded the real environment and virtual environment as a closed continuum, as depicted in Figure 1. From Figure 1, the far left is a purely real environment, and the far right is a purely virtual environment; everything in between is mixed reality, in which both real and virtual objects are present. The position of a user interface in this closed set is determined by the number of computer-generated elements in the user environment. In Figure 1, the far left represents a user interface entirely in the real physical world, whereas the far right represents a virtual environment entirely generated by computer. Between the extremes are AR, which refers to a real world augmented by virtual items, and augmented virtuality, which refers to an immersive virtual environment with elements from the real world.

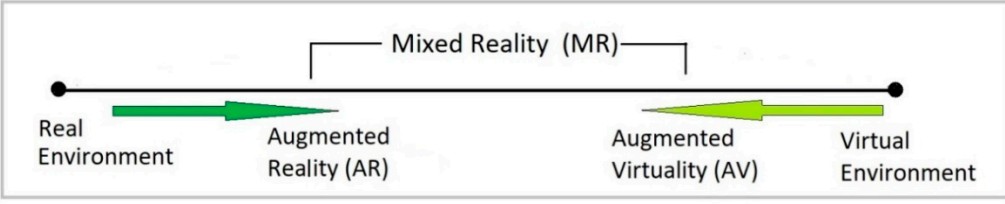

**Figure 1.** Reality–virtuality continuum. (Source: Milgram & Kishino's [9]).

In addition, Azuma [10] defined AR as an environment containing both virtual and real-world elements, and this definition is widely accepted by academics. By this definition, AR encompasses real and virtual items, real-time interaction, and a three-dimensional environment. By contrast, because VR fully immerses users in a computer-generated virtual environment, users cannot see the real environment around them. AR has been applied for guided tours of museums, outdoor tourist attractions, and school campuses,

enabling users to instantaneously acquire information regarding an educational or tourist venue. An image generated from AR requires the support of image recognition technology to integrate real and virtual objects. That is, the augmented content must be seamlessly superimposed on a real object so that the object can be vividly simulated on the display of a device. Thus, the AR application was designed to enable users to enjoy the fun of image scanning and recognition and thus afford them a unique visual experience and access to in-depth multimedia resources relating to the items on display.

With the change of the times, many educators are aware of the problems of the traditional teaching model; different from the traditional teaching method, teachers will begin to teach using situated learning [11,12]. Schools provide students with a learning situation. Teachers play a supporting role and stimulate students to actively participate in learning activities. Situated learning can help students develop autonomy, understand more difficult concepts and knowledge, and make students learn more meaningfully. The situated learning model combined with AR technology has been discussed in many studies [11–14] and has been applied in different learning fields. The research of Kamarinen et al. [15] pointed out that combining AR technology in a situated teaching way, students can have better learning interaction, and their understanding of knowledge principles is more in-depth than previous learning and can improve students' understanding of ecological and environmental science. In addition, the research results also point out that the situated teaching method can help students to apply the knowledge they have learned to the practical teaching field. Furthermore, in recent years, AR has developed rapidly, and research on the application of AR in the field of learning has also been published and widely applied to astronomy [16–18], mathematics [19,20], science [21,22], STEM education [2,4], and environmental education [3,23–26]. Furthermore, many researchers have pointed out that the application of AR in education has feasibility and positive benefits. Chen and Tsai [5] believe that applying AR to various subjects can improve students' learning motivation and learning effectiveness; Radu's literature [27] documents that AR can help improve students' learning motivation and also promote learning interaction and cooperation among students; it is identified that the feasibility and prevalence of AR as a teaching tool can assist teachers in teaching and have a positive impact on student learning. (See Bacca et al. [28]; Akçayır et al. [29]) Based on these studies, it can be assumed that AR technology has been effectively applied in education and has been effective in sharpening learners' perspectives and learning outcomes.

*2.2. Chatbot Technology*

2.2.1. Chatbot Background

Chatbot is an abbreviation of chat and robot. It uses natural language for input and simulates interpersonal conversation or chat interaction as output software through artificial intelligence [30]. A chatbot uses software-built commands to carry out voice conversations or text messages. ELIZA, developed by Joseph Weizenbaum at Massachusetts Institute of Technology (MIT) in 1966, was the world's first chatbot [31]. This product was designed in natural language and is now recognized as one of the first Turing-tested chatbots in the world. The basic mechanism of the system is to search the database for the corresponding keywords in the user's question, then match the keywords with the corresponding pattern, and finally output the matching answer. ELIZA is mainly used in the medical field. Through this system, mental patients can be guided to interact with the system, so as to obtain the information needed for treatment, and can provide auxiliary treatment for mental patients [31,32]. The software learns to have human-like emotions and simulates a situational conversation between a psychiatrist and a patient [33]. Since then, the ELIZA system has opened the era of intelligent chatbots. With the continuous increase of technological innovation and market demand, artificial intelligence is becoming an important bridge for data and technology to enter human life. In recent years, artificial intelligence technologies, including natural language processing, deep learning, speech recognition, and pattern recognition, have made steady progress, which has promoted the

high development of chatbot systems, and the method of human–computer interaction has also undergone great changes. These research contributions have formally ushered in the initial idea of chatbots operating through artificial intelligence, creating a critical basis for information science to enter the era of machine learning and tackling problems in life.

The application of chat robots has gradually become the direction of the development of various service platforms. By utilizing the interface base of chats and communications, it is possible to introduce a wide range of integrated services of different types and even to develop uncharted territories and topics. If the communication software can be applied to a variety of service platforms, it will attract more companies and software engineers to adopt and develop chatbots, increase the stability and ease of use, and create more stickiness, so that the number of chatbots and applications developed will be on a larger scale. In recent years, with the rapid development of artificial intelligence technology and the high usage rate of communication software, instant chatbots are the most commonly used artificial intelligence for our daily life. They appear as customer service agents, providing news, weather, navigation, and other information, as well as services, such as room reservation, ticket booking, shopping on line, and ordering a meal, etc. Furthermore, in addition to providing users with 24 h service throughout the year and saving a lot of labor costs for enterprises, it is easier and more inexpensive to be developed than websites and APP. For example, the Finplex chatbot in finance can accumulate knowledge through machine learning and can automatically sell products and answer customer questions [34]; in addition, during the COVID-19 epidemic, Battineni et al. [35] developed chatbots to talk to patients and understand their psychological conditions, conduct medical diagnostic assessments, and recommend actions to be taken. According to Business Insider (2016), about 3 billion people in the world use mobile messaging applications. Chatbots reach more people through communication platforms and are regarded as a way to directly interact with users or customers [36].

### 2.2.2. Types of Chatbots

Nimavat and Champaneria [37] classify chatbots based on knowledge domain, service provided, goal, input processing, and response generation method; however, a chatbot does not necessarily belong to a certain type, and it may be a mixture of multiple types. Among the general classifications, the knowledge-based classification can be subdivided into a closed domain and an open domain. Closed-domain chatbots usually focus on answering specific domains and cannot answer other questions outside the domain; open-domain chatbots can talk about anything and provide appropriate responses. When selecting a model for development, it is important to understand the definition of a chatbot service framework. In the open domain, there are no restrictions on topics, and user conversations do not necessarily have explicit instructions. This open framework, which is usually trained through data processed by machine learning, requires the accumulation of numerous topics or a huge amount of knowledge before the service can respond to the user's expectations, such as Siri, Apple's personal assistant, or Google Assistant on smart devices.

On the other hand, closed domains are mostly closed responses that restrict the user's input functions and objectives. The system is more specific in its subject matter, although it is limited and restricts the user's ability to answer questions, with a bias towards completing as many subject-specific tasks as possible, in such application scenarios as Internet banking, food ordering, airline ticketing, entertainment, and education.

Furthermore, chatbots classified by input processing and response generation can be divided into three types:

1. Rule-based model: Currently the most common type is a rule-based closed chatbot. This chatbot replies to the edited text by recognizing the input vocabulary and does not generate a new answer by itself; this method is based on the sentence inputted by the user, finding the matching question template in the template library, and then generates the answer according to the corresponding answer template; for example, ALICE uses the AIML language [38] to describe the knowledge database, and AIML

uses XML syntax to store data. The advantage of this approach is that it is accurate, but the disadvantage is that it requires a lot of manpower, lacks flexibility, and has poor scalability.

2.  Retrieval-based model: Based on retrieval technology, the chatbot matches the sentence entered by the user in the dialogue database by using the application programming interface (abbreviated API) and other technologies to query and utilize existing data. The system gives the user an appropriate answer to the question by means of an algorithm. This model is more closed, and the program is designed to generate fixed content in advance.

3.  Generative model: This model does not pre-construct content data; it learns from scratch, generating new answers from each question, and, through this application, machine learning translation techniques are used to translate the content input to the user after deep learning. This type of chatbot understands the input content through natural language, and only through the deep learning of huge data can a good dialogue be achieved.

### 2.2.3. Chatbots for Education

Chatbots have been successfully used in various fields [33], including legal, military, business, and education. The application of chatbots in education can be used as an auxiliary learning tool and has become a trend in recent years [39]. There have been many educational applications of chatbots in recent years [40–47], showing that chatbots have the potential to improve learning outcomes [48]; however, the use of chatbots to assist teaching is still in its early stages [39]. By designing course topics into the chatbot, students can no longer only absorb the knowledge in the classroom one-sidedly, and the chatbot combined with mobile devices as an auxiliary teaching tool can also ensure more diverse teaching. Moreover, it has changed the traditional education method, made up for the lack of teachers' teaching, and can provide a wider range of teaching materials for students to learn.

Chen et al. [40] use chatbots for Chinese single-word learning, research results indicate that chatbots can improve students' performance, and that using chatbots in one-on-one personal tutoring can achieve better learning outcomes than using chatbots in many-to-one classrooms. Pham et al. [46] designed a chatbot to assist users in practicing English; its functions include general greetings, responding to users' specific requests, prompting or explaining learning content, sending reminders to users, etc. The chatbot can talk to students in a pleasant atmosphere, effectively attracting students' attention. In addition, chatbots can also help students with difficulties to make rapid progress in their studies, narrowing the gap between minority and mainstream students, and adjusting to different cultures and environments; suitable chatbots can bring positive effects to students influences [40]. Hobert and Meyer [47] designed a chatbot as a teaching assistant for programming teaching. This chatbot can answer users' open-ended questions, automatically evaluate the programs submitted by users, and use natural language to guide users to complete exercises. Bailey et al. [41] developed a chatbot that can interact with students and share story content to help language learning. The experimental results show that chatbot interaction is positively correlated with self-confidence and that students with high self-confidence are willing to work hard and spend more time for study. Nerdy Bot, developed by Nerdify, aims at the problems faced by students, making learning easier and more efficient [42]. The software utilizes Facebook Messenger to instantly answer students' homework-related questions to speed up learning. Xu et al. [43] developed a robot that reads with children. The chatbot can expand and extend more sentences according to the children's dialogue training. The experimental results show that a robot that talks in a guided manner improves children's reading comprehension. Therefore, children can express sentences that are easier to understand after talking with the chatbot. Hwang and Chang's research [44] mentioned that the use of chatbot education in the past mostly used guided strategies, and it is suggested that it can be designed for different learning activities, so that students can design for different

learning activities. Moreover, students can not only complete tasks but also use interact with classmates to create a more interactive learning space.

### 2.2.4. Chatbots during COVID-19 in Taiwan

As the COVID-19 pandemic rages across the globe in 2019, many countries are under total lockdown and people have to stay at home to protect themselves from the disease, relying on communication software to provide real-time information. With extended periods of time at home, chatbots play a crucial role in providing services. According to PSFK research [49], 74% of consumers preferred to find immediate answers on chatbots, and companies using chatbots in the retail business were seen as efficient (47%), innovative (40%), and helpful (36%). The use of chatbots with AI allows companies to better understand user behavior, habits, and preferences and to provide a 24 h service that can handle users from different regions in a multi-tasking manner. By reducing the amount of time people spend in contact with each other through the Internet, the chatbot service offers efficient and convenient applications, shifting from the basic single function of the past to a diverse and professional application scenario.

Taiwan's Ministry of Health and Welfare has launched a Messenger chatbot, developed with the help of GoSky, to ensure that the public has access to the latest and correct information on COVID-19. The automated chatbot will provide citizens with information about COVID-19. In addition to providing updates, the chatbot will also be used in the CDC's Facebook Live to provide general public COVID-19-related content. Moreover, the Central Epidemic Command Center actively promotes various scientific and technological epidemic prevention measures. Among them, the Google Assistant service is used to build an "epidemic prevention expert" intelligent chatbot in Chinese and English, so that both Chinese and foreigners in Taiwan can use multiple channels to obtain the latest epidemic prevention information. Google Assistant provides a variety of real-time epidemic prevention information in the form of questions and answers, such as: disease introduction, transmission route, clinical symptoms, data on confirmed and dead cases in various countries, travel epidemic advice, what to do after suspected infection, community epidemic prevention actions, home isolation, and home quarantine instructions and information on purchasing measures for masks.

Since the success of an e-learning system depends on students' willingness to adopt and acceptance of this system [50]. At present, students generally carry smartphones with them, and the smartphones are equipped with chatbots as mobile learning vehicles. They are not limited by time and space, can answer questions in real time, and have the characteristics of humanized interaction. These functions and services can just meet the learning needs of the new generation of nursing staff. Chuang et al. [51] created a chatbot teaching program designed on the social media LINE, so that nursing educators can integrate technology into clinical teaching, develop multiple innovative teaching materials, and create a learning environment to promote professional learning and facilitate self-growth in nursing staff. In addition, with mobile device carriers, not only can it be diversified and high-quality communication and transmission effects be achieved, ordinary people can easily install and download the free-to-use LINE software to maintain interpersonal interaction and communication. Thus Chi [52] proposes the effectiveness of chatbot counseling for pregnant women during COVID-19, by investigating cases and collecting the behavioral experience of chatbots during pregnancy and the impact of the new crown pneumonia epidemic on the choice of counseling channels. It can be used as a reference for the clinical development and design of such nursing guidance chatbots or health consultation promotion in the future. Furthermore, due to the epidemic, couples cannot directly contact manufacturers, which has caused a huge impact on practitioners in the wedding industry. Through project management and the concept of chatbots, Nian [53] designed a wedding consultant platform that interacts with couples at any time and combines intelligent customer service chatbots. This chatbot can answer the couple's questions, assist the couple in planning their

wedding, and provide suggestions for the company to prepare the next marketing strategy and product development direction.

It is clear from the above services that the current pandemic crisis has not subsided and that many teams are utilizing technology to develop related chatbots, using task-oriented and closed-domain types of services, with the aim of addressing people's lives and generating knowledge-dissemination effects. This research therefore seeks to create a task-oriented, professional, and user-friendly chatbot. Further, the authors wanted to design an application interface that was easy to use, easy to learn, and increased in interactivity, so AR technology was leveraged to meet the content developed for this product. This study therefore examines the technical analysis of the chatbot platform and the use of the Messenger core, retrieval model, and closed domain for chatbots with AR technology in an educational context.

### 2.3. ARCS Model

The ARCS model was proposed by Keller [54,55], integrating many motivational theories. The purpose of the models is to aid in curriculum design or to improve teacher teaching. Keller believes that this motivational design model can be applied to learners of all ages and makes the design of teaching materials more suitable for motivating learners' participation and interaction. He also believes that, if the teaching material developed by any kind of instructional design cannot arouse the interest or concentration of learners, the effect of learning will be greatly reduced. Therefore, Keller expects that the ARCS model can provide educators with a design strategy for identifying and understanding teaching according to students' motivational needs, so as to stimulate learning motivation and effectively improve students' learning and performance.

The main components of the ARCS model are: "Attention", "Relevance", "Confidence", and "Satisfaction". The process of this mode is mainly to arouse students' attention and interest in the course, then let them find that such learning is closely related to their life, and finally let students have enough ability and confidence to deal with it, and students will obtain learning achievement and satisfaction. Furthermore, the focus of the motivation model is on the internal factors of the learner (such as personal value, expectation, ability, and cognitive value, etc.), and the external factors of the teaching environment (such as cooperation and the planning of the teaching design).

1.  "Attention": The first step in this mode is to get the student's attention. If students do not have considerable attention and interest in a subject, learning will not be effective. It is not difficult to obtain students' attention, but the real challenge lies in how to keep students' attention and interest in the course. In addition, it is also necessary to consider the use of various design strategies in teaching materials to maintain the freshness of students' knowledge.

2.  "Relevance": The second element in the model is to give students a relevant awareness of learning. Although new things can help to concentrate attention, people often tend to combine the knowledge they are already familiar with and understand for task-based learning. Therefore, the design that conforms to the characteristics, knowledge, and cultural background of students is an indispensable prerequisite for improving students' interest in learning. In addition, teachers can make good use of skills to persuade students that this course is related to future life and career, that is, learners must also be aware that their personal needs are met by teaching. Therefore, teaching must meet students' goals, let students know the advantages of participating in teaching activities, properly grasp the sense of familiarity, connect students' previous experience, and arouse students' learning motivation.

3.  "Confidence": Confidence is related to students' expectations of success or failure and affects students' actual effort and performance. After successfully arousing students' attention and relevance, if the teacher ignores the students' fear of a certain subject and thinks it is too difficult or if the content is not challenging and too simple, both of these will reduce the students' learning motivation and learning outcomes. Therefore, in the

teaching plan, teachers should design courses that match the individual abilities of students, assist everyone to achieve success, and ensure their confidence in continuing to learn.

4. "Satisfaction": Satisfaction is an evaluation of students' learning results, and personal satisfaction is an important factor for motivation to continue. The most direct way is to allow students to apply the knowledge concepts they have learned to the environment in the form of self-expression. Therefore, teachers should maintain fairness in teaching, pay attention to whether the initial goal of the course is consistent with the results of what students have learned, and provide contextualized learning so that students can experience the satisfaction of applying what they have learned.

The four factors are interlocking and affect the learning effect of students. Teachers must make their teaching conform to the ARCS model, so that students' learning can produce a virtuous cycle. If any link is lacking, the overall teaching effect will be greatly reduced. Keller [54,55] emphasized that, if learners are deficient in these four conditions, teachers can implement systematic teaching strategies for students' deficiencies, increase students' motivation for deficiencies, and improve learning effects.

## 3. Research Model and Hypotheses

In order to understand the effectiveness of the system when users operate the chatbot built in this study, a questionnaire method was employed to investigate students' attitudes towards learning the natural science by adding questions to the basic data in the first stage. Students' attitudes towards learning are influenced by the situations or experiences in the learning process. Further, this study aims to incorporate students' attitudes towards learning the natural subject at school as a reference for the improvement of the system. The second stage of the questionnaire was designed with reference to the literature on the ARCS motivation model to find out what the learning indicators were after the students had operated the system.

### 3.1. Hypothesis Development

The four elements of the ARCS model must be matched with each other to induce learning motivation. Our research hypothesis is based on Keller's ARCS model [54,55] and refers to the hypotheses of Chang et al. [56,57] and Lin et al. [58]. Figure 2 illustrates the study's theoretical model. The hypotheses tested in the model include the following:

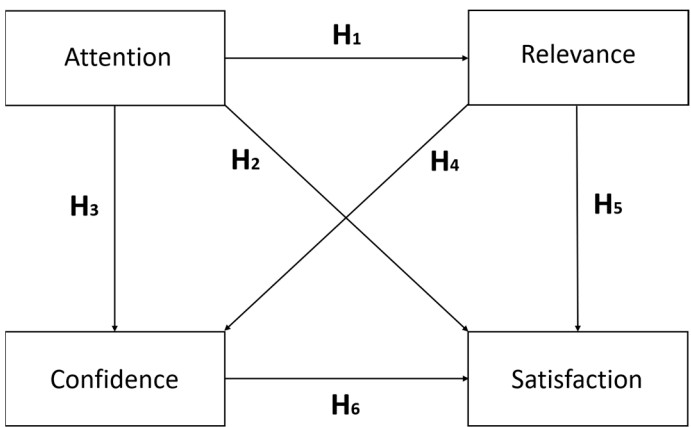

**Figure 2.** The theoretical model [55,56].

**Hypothesis 1 (H1).** *"Attention" predicts "Relevance".*

**Hypothesis 2 (H2).** *"Attention" predicts "Satisfaction".*

**Hypothesis 3 (H3).** *"Attention" predicts "Confidence".*

**Hypothesis 4 (H4).** *"Relevance" predicts "Confidence".*

**Hypothesis 5 (H5).** *"Relevance" predicts "Satisfaction".*

**Hypothesis 6 (H6).** *"Confidence" predicts "Satisfaction".*

*3.2. ARCS Model's Construction*

The questionnaire was designed by the researcher by referring to the literature on the ARCS motivational model and was integrated into the ARCS model of the biology subject by means of information technology tools. The ARCS motivational model was developed by educational psychologist Keller [54,55] with the aim of finding out whether students were motivated to learn when using the online system and to understand the indicators of improvement in the use of the system's teaching materials through the construction of a chatbot. The ARCS model is made up of four constructs: "Attention", "Relevance", "Confidence", and "Satisfaction". The second phase of the improvement survey in this study relied on these four approaches to understand how the students paid attention to learning, felt relevant when using the software, and were able to build confidence and gain satisfaction for planning the software.

1.  "Attention": In designing the platform, the researcher recognizes that, if the topic does not engage students' interest, then the software learning will affect their level of attention during the learning process, so being able to draw attention to the learning is an essential strategy for system planning. Students' curiosity in learning can be stimulated and reinforced through the following approaches:
    (1) In designing the AR chatbot curriculum, the use of online digital media was enhanced by searching the web for materials that could be circulated and modified to match the visually appealing animated video content and images, as well as incorporating the teaching content and introduction to the creature species.
    (2) The AR chatbot interface was designed to be easy to understand, easy to operate, and more logical.

2.  "Relevance": Through the design of the biology content, students were able to generate cognitive resonance, connect knowledge and relate experiences to their previous textbook learning in the classroom, and master their biology learning priorities:
    (1) When designing the AR chatbot curriculum, biological knowledge was added so that the content could be linked to everyday life and students could learn about nature without having to go outside.
    (2) The AR chatbot course was designed to cover the meaning of higher education-oriented online learning.
    (3) When designing the AR chatbot course, the content was constructed to fit into the systematic teaching curriculum categories.

3.  "Confidence": After completing the content, students were able to form effective links to reduce their fear of learning and increase their confidence through online self-directed learning.
    (1) Course content was designed to be moderately difficult to enhance students' confidence in learning.
    (2) Students were given continuous self-management of their time and online learning using the chatbot.

4.  "Satisfaction": Feedback on learning with the software was given, and what was learned was put to use, effectively passed on, and shared with others.
    (1) Feedback was given on peer learning with chatbots.
    (2) The opportunity was available to apply knowledge in the classroom, in higher education examinations, and in life.

## 4. Methodology

The authors constructed an AR-based chatbot system. In order to illustrate the learning effect of users operating the system, we used the questionnaire survey method to verify whether the method proposed in this study can arouse students' interest in learning. The measurement tools of this research include questionnaires on the experience of junior high school students studying biology and the ARCS learning motivation questionnaire.

### 4.1. The AR-Based Chatbot System

In this study, we used the Chatfuel platform [59] to build a chatbot and linked to an AR HTML webpage written through the Adobe Brackets [60] compiler. The content used 3D models of living creatures from the Sketchfab platform [61] and complied with the conditions of Creative Commons 4.0 to download images in USDZ format to present the appearance of the creatures, allowing students to manipulate the AR creature application and understand the morphological characteristics and distribution of the creatures through the compilation of textual descriptions and video teaching contents of the creatures. The system architecture is shown in Figure 3.

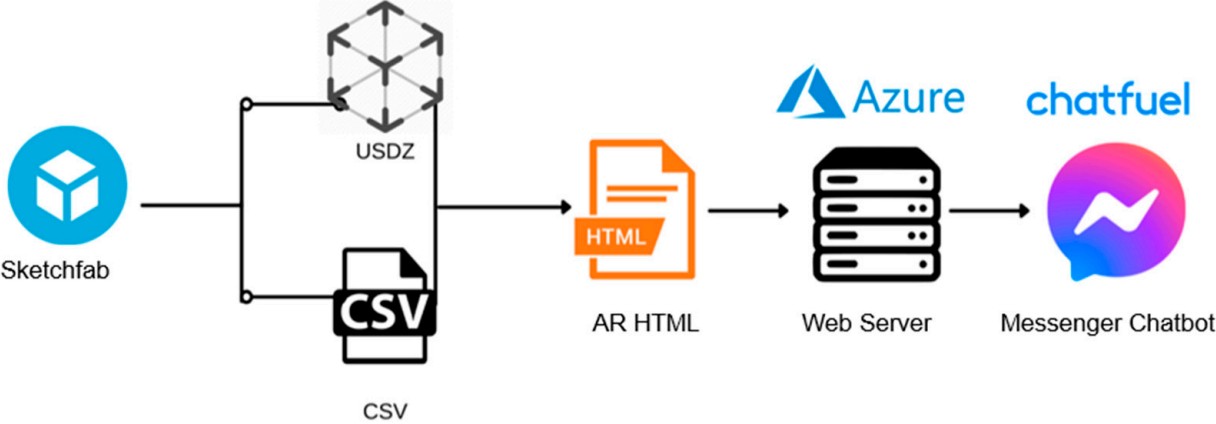

**Figure 3.** The AR-based chatbot system architecture.

Messenger is a communication software service, a product of Facebook, which features social chatting and the online display of corporate shops and product after-sales consultation and is widely used in desktop devices, web pages, mobile applications, on-board units, etc. Currently, users can search for a very wide range of chatbot online services, which are used in different application scenarios. Compared to other mobile applications, Messenger allows developers to develop the content of the service without having to write applications for both Android and iOS platforms, with a smooth functional interface that enables developers to focus on the content of the service. In addition to the number of users and the diversity of software applications, a chatbot service is now available on Messenger, allowing users to apply for a fan page online through their personal or business accounts. In contrast to personal pages, official fan pages are available for the public, such as businesses, educational organizations, celebrities, corporate entities, etc. One can apply for fan pages according to different types of needs. When programming, Facebook offers the Messenger Bot API, a functional module for chatbots, for page design and messaging functions. With support for functional testing, backend permissions can be graded at different stages to ensure application by users with different identities. The fan page supports button-driven chatbots with services related to feature testing, ad pushing interaction, sharing messages and communication, and relationship maintenance.

It is clear from the above services that the current pandemic crisis has not subsided and that many teams are utilizing technology to develop related chatbots, using task-oriented and closed domain types of services, with the aim of addressing people's lives and generating knowledge dissemination effects. We therefore seek to create a task-oriented,

professional, and user-friendly chatbot. Therefore, the authors design an application interface that was easy to use, easy to learn, and increased in interactivity, so AR technology was leveraged to meet the content developed for this product. This study therefore examines the technical analysis of the chatbot platform and the use of the Messenger core, retrieval model, and closed domain for chatbots with AR technology for education.

For this research, we applied some AR technology and messenger chatbot to develop our system. We developed the system for mobile devices and web-based systems, to be able to expose it to students studying biology to enjoy this interactive experience and enhance their learning experience. The software used to develop the application is listed below:

- Chatfuel [59]: a self-serve platform for building Facebook Messenger chatbots. The platform has an intuitive visual interface that build chatbot flows and establishes conversational rules;
- Adobe Brackets [60]: a source code editor with a primary focus on web development;
- Sketchfab [61]: a 3D modeling platform website to publish, share, discover, buy, and sell 3D, VR, and AR content;
- Canva [62]: a graphic design platform used to create social media graphics;
- Microsoft Azure [63]: a cloud computing platform operated by Microsoft for application management.

From the perspective of user-centered software design, since the user is a junior high school student, the interface design is developed with simplicity and rich colors. The steps we developed are described in more detail in Table 1.

**Table 1.** Construction steps of AR-based chatbot system.

| Steps | Demonstration |
|---|---|
| Step 1:<br>Construct Chatfuel related settings. The process of constructing a dialogue includes the user's welcome screen message, usage rules, course structure, and AR overview page, so that learning users can understand how to operate. | 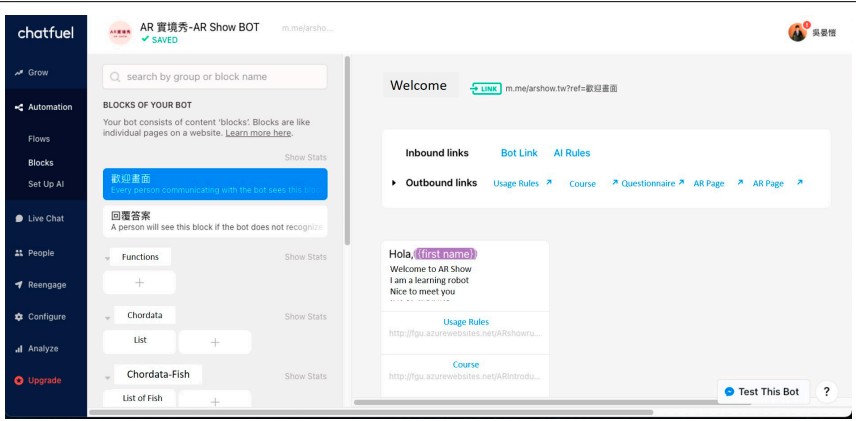 |
| Step 2:<br>Graphical interface menu design. Use Canva to design pictures and use brighter colors. It is hoped that students can clearly operate at first glance and increase their attention during learning. | 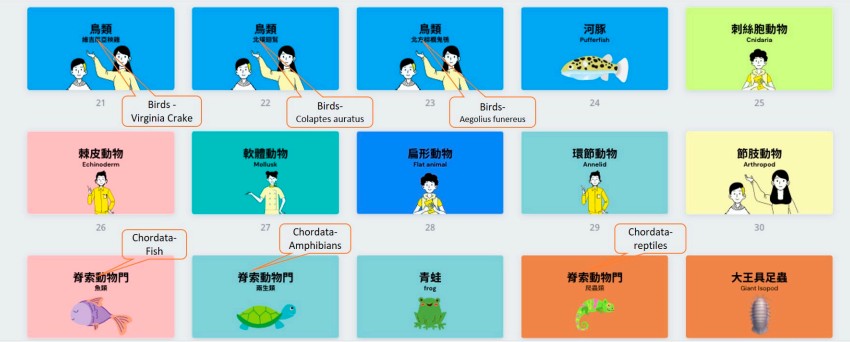 |

**Table 1.** *Cont.*

| Steps | Demonstration |
|---|---|
| Step 3:<br>Create a biological 3D model on the Sketchfab platform.<br>Students can establish a link with relevant knowledge of biology, immediately associate with the natural knowledge they have learned, and achieve real-time interactive learning effects, helping students understand the appearance of creatures. | 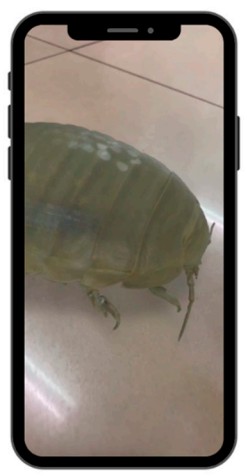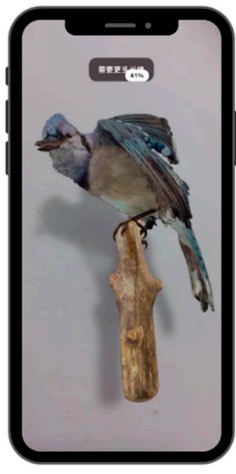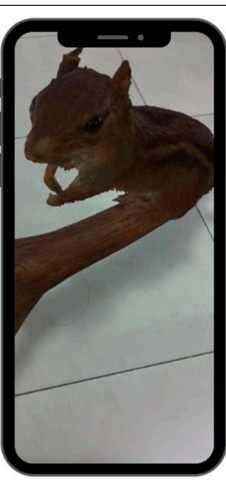 |
| Step 4:<br>Apply for a messenger chatbot. (https://www.facebook.com/arshow.tw/, accessed on 2 November 2022).<br>Students can establish a link with relevant knowledge of biology, immediately associated with the natural knowledge they have learned. | 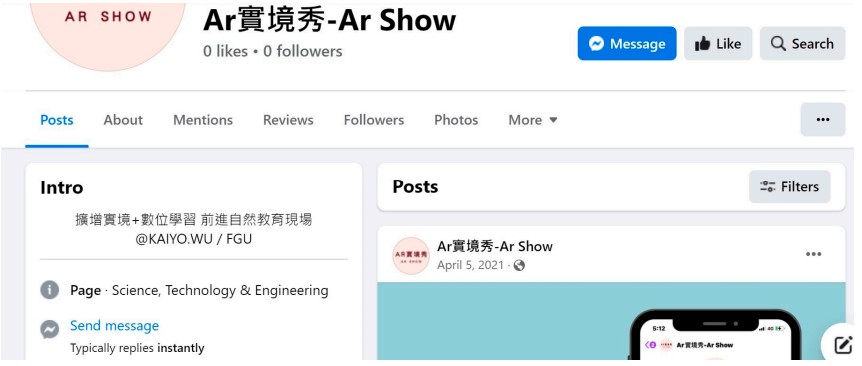 |

### 4.2. Experimental Procedure

To assess the proposed model presented in Figure 2, the target participants for the project were the students in a junior high school in Yilan county, Taiwan. In particular, a questionnaire was administered to 109 students in four classes of first-year students of a junior high school. The total number of questionnaires returned after the distribution of the questionnaires was 109, with 102 valid questionnaires and a return rate of 93%. Students were asked to spend their time at home after school on the learning operations based on the teaching objectives and learning contents of the biology class, including the definition and introduction of the species of cnidarians, platyhelminths, molluscs, annelids, echinoderms, chordates, fish, amphibians, reptiles, birds, mammals, etc. A questionnaire comprising two sheets was distributed during the biology class. The students completed the questionnaires and returned them to the biology teacher for consolidation, coding, and analysis. Further, the web page was designed in a responsive manner so that users could learn the content directly on the one-page web page after the chatbot had pressed a button. The screen was designed in such a way that the main learning content could be centered and so the content and text size could be adjusted such that students could focus their attention while learning and would be less likely to be distracted by other influences. This questionnaire was divided into two stages with a total of 40 questions, of which the first stage was for basic personal information and the average time spent on the study of the biology at home in a day, by referring to the students' attitude towards learning the subject, which included three indicators (learning ability, application ability, and problem-solving ability) to check whether the students had experience in using chatbots, with 20 items in total. The second stage used the four indicators from the ARCS motivational model as indicators for system improvement, with a total of 20 items.

## 5. Results

The total number of questionnaires was 109. After excluding invalid questionnaires, 102 valid questionnaires were obtained, including 48 males (47.1%) and 54 females (52.9%). The first part of this survey includes gender, the average time spent studying biology at home, the experience of using chatbots, and the three dimensions of attitudes towards learning biology (learning ability, application ability, problem-solving ability); see Table A1 for details. Furthermore, the questionnaire results of the four indicators in the ARCS model are also shown in Table A2. After the questionnaire data was collected and performed by IBM's Statistical Product and Services Solution (SPSS) V.26, these were statistically analyzed:

1. Questionnaire validity and reliability analysis for the ARCS model;
2. One-way ANOVA test and Scheffe's method to determine whether the external variables affect the four indicators ("Attention", "Relevance", "Confidence", and "Satisfaction") of the ARCS model;
3. Pearson's correlation analysis of the hypotheses for the ARCS model;
4. Regression analysis to validate research hypotheses.

### 5.1. Scale Validity and Reliability

#### 5.1.1. Validity Analysis

The Kaiser–Meyer–Olkin (KMO) metric is used to measure the suitability of data for factor analysis. KMO measures the adequacy of sampling for each variable in the test model and in the overall model. A statistic is a measure of the proportion of variance between variables that may be ordinary variance. The lower the proportion of variance, the better your data are for factor analysis. A KMO metrics return values between 0 and 1. A rule of thumb for interpreting statistics: a KMO measure between 0.8 and 1 indicates adequate sampling. A KMO measurement of less than 0.6 indicates insufficiency and that remedial action should be taken [64]. Bartlett's test of sphericity tests the assumption that the correlation matrix is an identity matrix, which indicates that the variables are uncorrelated and therefore not suitable for the testing of structure. Small values of significance levels (less than 0.05) indicate that factor analysis may be useful for your data [65,66].

In this study, the KMO and Bartlett's spherical test were used to determine the suitability of the factor analysis. Table 2 shows the KMO and Bartlett's sphericity test results. The measure of the adequacy of KMO sampling was between 0.8 and 0.9, indicating that the sample size is large enough to evaluate the factor structure, while the Bartlett's spherical test value was less than 1% of the significance level. As a result, "Attention", "Relevance", "Confidence", and "Satisfaction" were all suited for the factor analysis.

**Table 2.** KMO and Bartlett's sphericity test results.

| | | "Attention" | "Relevance" | "Confidence" | "Satisfaction" |
|---|---|---|---|---|---|
| KMO value of sampling adequacy | | 0.847 | 0.852 | 0.827 | 0.872 |
| Bartlett's Sphericity Test | Approx. Chi-Square | 286.830 | 230.220 | 248.414 | 387.572 |
| | Degree of freedom | 10 | 10 | 10 | 10 |
| | Significance | 0.000 | 0.000 | 0.000 | 0.000 |

#### 5.1.2. Reliability Analysis

The Cronbach's alpha coefficient is a measure of scale reliability and is a convenient test used to estimate how closely related a set of items are as a group. Theoretically, Cronbach's alpha results should give you a number from 0 to 1. The general rule of thumb is that a Cronbach's alpha of 0.70 and above is good, 0.80 and above is better, and 0.90 and above is best. Higher reliability indicates that measurement errors are smaller. In this research, Cronbach's $\alpha$ is calculated to determine the consistency of the "Attention", "Relevance", "Confidence", and "Satisfaction" scale and is found to be 0.889, 0.862, 0.875, and 0.936,

respectively, indicating that the reliability of the question items is very high. That is, from Table 3, the questionnaires of this study measured consistency in these four constructs.

**Table 3.** Cronbach's analysis for variables.

| Constructs | Cronbach's Alpha |
| --- | --- |
| "Attention" | 0.889 |
| "Relevance" | 0.862 |
| "Confidence" | 0.875 |
| "Satisfaction" | 0.926 |

*5.2. Single Factor Covariace Analysis: One-Way ANOVA Test and Scheffe's/LSD Method*

When more than three groups exist, ANOVA can be used to determine whether a variable differs significantly between the groups. In this study, we want to check whether there is significance in the amount of time students spent studying biology, and the scale dimensions of ARCS model was examined using one-way ANOVA. That is, we employed ANOVA to determine whether external variables ("How much time do you spend on studying biology lessons per day?") had significant effects on "Attention", "Relevance", "Confidence", and "Satisfaction". We employed ANOVA to examine this variable and the variance of the four dimensions; only the "Confidence" variable was found to be statistically significant, according to Table 4.

**Table 4.** One way ANOVA variance results.

| Constructs | Source | Sum of Squares | df | Mean Square | F Value | p |
| --- | --- | --- | --- | --- | --- | --- |
| "Attention" | Difference between groups | 28.358 | 2 | 14.179 | 1.063 | 0.349 |
| | | 1320.397 | 99 | 13.337 | | |
| "Relevance" | Difference between groups | 45.812 | 2 | 22.906 | 1.897 | 0.155 |
| | | 1195.600 | 99 | 12.077 | | |
| "Confidence" | Difference between groups | 105.518 | 2 | 52.759 | 3.916 | 0.023 * |
| | | 1333.659 | 99 | 13.471 | | |
| "Satisfaction" | Difference between groups | 69.955 | 2 | 34.978 | 2.228 | 0.113 |
| | | 1554.006 | 99 | 15.697 | | |

Variables of significance (* $p \leq 0.05$).

Scheffe's and LSD methods are post-hoc tests used in the analysis of variance. After we have run ANOVA and obtained a significant F-statistic, we can run Sheffe's or the LSD test to find out which pairs of means are significant. Scheffe's test corrects alpha for simple and complex mean comparisons. Complex mean comparisons involve comparing more than one pair of means simultaneously. Therefore, in this study, Scheffe's method and LSD were adopted for comparison, and analysis was carried out based on the time spent studying biology. Table 5 shows that the three groups of studying time were compared and the results were found to be correlated at a significant level after analysis. It was concluded that the amount of time students spent on studying biology was effective in increasing their confidence when using the chatbot system. It is possible to realize that learning skills are reinforced through manipulation and to feel that learning through the system strengthens the application of knowledge in everyday life and helps when exploring the user interface. In addition, in exploring the software interface, students were willing to operate the chatbot and had the confidence to teach students who did not know how to use digital online learning to try to learn the course well, so as to achieve the effect of promoting student learning. The above analysis of this study reveals the following:

**Table 5.** The Scheffé method results for "How much time do you spend on studying biology lessons per day?".

| Constructs | Method | I (h) | J (h) | Mean Difference (I-J) | Standard Error | $p$ | 95% Confidence Interval | |
|---|---|---|---|---|---|---|---|---|
| | | | | | | | Lower Bound | Upper Bound |
| "Confidence" | Scheffé method | 0.5–1 | 1–1.5 | −1.72308 | 0.85169 | 0.135 | −3.8397 | 0.3936 |
| | | | >1.5 | −2.75804 | 1.19663 | 0.075 | −5.7320 | 0.2159 |
| | | 1–1.5 | 0.5–1 | 1.72308 | 0.85169 | 0.135 | −0.3936 | 3.8397 |
| | | | >1.5 | −1.03497 | 1.32015 | 0.736 | −4.3159 | 2.2459 |
| | | >1.5 | 0.5–1 | 2.75804 | 1.19663 | 0.075 | −0.2159 | 5.7320 |
| | | | 1–1.5 | 1.03497 | 1.32015 | 0.736 | −2.2459 | 4.3159 |
| "Confidence" | LSD method | 0.5–1 | 1–1.5 | −1.72308 | 0.85169 | 0.046 * | −3.4130 | −0.0331 |
| | | | >1.5 | −2.75804 | 1.19663 | 0.023 * | −5.1324 | −0.3837 |
| | | 1–1.5 | 0.5–1 | 1.72308 | 0.85169 | 0.046 * | 0.0331 | 3.4130 |
| | | | >1.5 | −1.03497 | 1.32015 | 0.435 | −3.6544 | 1.5845 |
| | | >1.5 | 0.5–1 | 2.75804 | 1.19663 | 0.023 * | 0.3837 | 5.1324 |
| | | | 1–1.5 | 1.03497 | 1.32015 | 0.435 | −1.5845 | 3.6544 |

Variables of significance (* $p \leq 0.05$).

In the one-way ANOVA, there was a significant difference between the time spent on studying from Table 4. The LSD analysis revealed a considerable difference between 1.5 h or more of study time and 0.5–1 h (I-J), shown in Table 5, with the data being positively correlated, and no overestimation can be made for the remaining time. The result of the comparison is that the chatbot platform is an online learning tool, and the length of time spent studying is a factor in boosting confidence in the digital teaching tool. The length of time measured has led to the initial results that the duration of time spent studying biology is a key influence on the indicators.

*5.3. Pearson's Correlation Analysis*

This section is a cross-check of the systematic teaching material improvement scale dimensions by Pearson, with a Pearson correlation coefficient of 0.7 or above being highly relevant. Discriminant validity is tested by Pearson's correlation value. Pearson's correlation coefficients are calculated to determine the correlations between the following variables: "Attention", "Relevance", "Confidence", and "Satisfaction". The Pearson Correlation coefficient is a measure of linear correlation between two sets of data. The correlations in the table below are interpreted in the same way as those above, such that the result always has a value between −1 and 1, with −1 indicating a perfect negative correlation, +1 indicating a perfect positive correlation, and 0 indicating no correlation at all. Pearson's correlation coefficients were calculated to determine the correlations between variables. The analytical results as depicted in Table 6, and Figure 4 indicated strong correlations between four main constructs. The results show that every pair of the variables are significantly correlated at the level of 0.01. Additionally, the constructs were highly correlated with each other. This verified that the main dimensions in the research framework were significantly correlated, verifying the ARCS model. Therefore, this study reveals the followings based on the results in Table 6:

According to Figure 4 and Table 6, **H1** assumes that, by operating the AR-based chatbot system, students will not only be interested in learning but will also be able to connect to the knowledge in the textbook and develop a positive attitude towards learning, so that drawing attention has a significant impact on "Relevance". The Pearson correlation coefficient of 0.812 indicates that the hypothesis (**H1**) is supported, and the results show a high degree of positive correlation.

**Table 6.** Pearson's correlation results.

|  |  | "Attention" | "Relevance" | "Confidence" | "Satisfaction" |
|---|---|---|---|---|---|
| "Attention" | Pearson correlation | 1 | 0.812 ** | 0.734 ** | 0.823 ** |
|  | Significance (2-tailed) |  | 0.000 | 0.000 | 0.000 |
|  | N | 102 | 102 | 102 | 102 |
| "Relevance" | Pearson correlation | 0.812 ** | 1 | 0.773 ** | 0.804 ** |
|  | Significance (2-tailed) | 0.000 *** |  | 0.000 | 0.000 |
|  | N | 102 | 102 | 102 | 102 |
| "Confidence" | Pearson correlation | 0.734 ** | 0.773 | 1 | 0.781 ** |
|  | Significance (2-tailed) | 0.000 | 0.000 |  | 0.000 |
|  | N | 102 | 102 | 102 | 102 |
| "Satisfaction" | Pearson correlation | 0.823 ** | 0.804 ** | 0.781 ** | 1 |
|  | Significance (2-tailed) | 0.000 | 0.000 | 0.000 |  |
|  | N | 102 | 102 | 102 | 102 |

Variables of significance (** $p \leq 0.01$, *** $p \leq 0.001$).

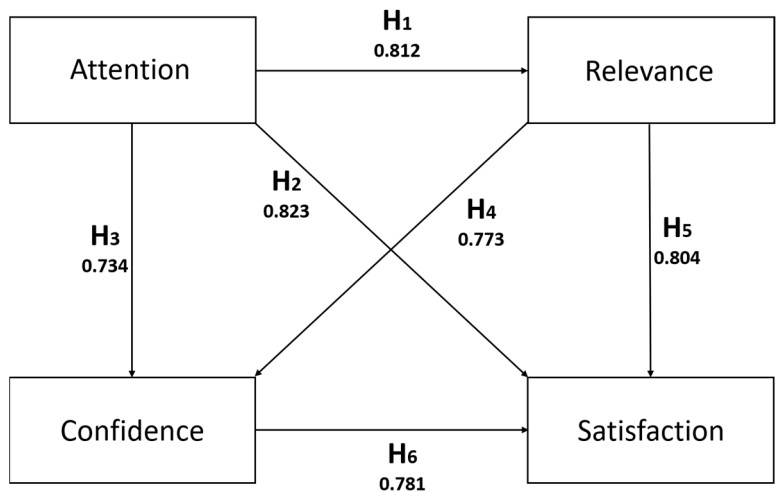

**Figure 4.** The Pearson's correlation results for this theoretical mode.

**H2** assumes that, by operating the AR-based chatbot system, students can enjoy learning and exploring and can gain a sense of achievement through learning in an enjoyable atmosphere, so that attention can have a significant impact on "Satisfaction", with the Pearson correlation coefficient of 0.823 indicating that the hypothesis (**H2**) is supported.

**H3** assumes that, by operating the AR-based chatbot system, students are motivated to learn and are given the opportunity to manage their own time and arrange their own online learning, which helps to build their confidence, so that attention can have a noticeable effect on "Confidence"-building, with the Pearson correlation coefficient of 0.734 indicating that the hypothesis (**H3**) is confirmed.

**H4** assumes that by operating the AR-based chatbot system, students will become familiar with the curriculum and make connections, which in turn will lead to feelings of relevance and increase confidence and willingness to learn, so that "Relevance" has a marked impact on "Confidence"-building, with the Pearson correlation coefficient of 0.773 indicating that the hypothesis (**H4**) is proved.

**H5** assumes that, by operating the AR-based chatbot system, students are able to get to grips with this new type of online learning and feel that it is worthwhile, so that "Relevance" has a marked effect on "Satisfaction", with the Pearson correlation coefficient of 0.804 indicating that the hypothesis (**H5**) is supported.

**H6** assumes that, by operating the AR-based chatbot system, students are able to command the important content and have confidence in acquiring knowledge, increasing their sense of achievement when learning, so that "Confidence"-building has a significant

impact on the satisfaction gained, with the Pearson correlation coefficient of 0.781 indicating that the hypothesis (**H6**) is supported.

*5.4. Hypotheses Model Test: Regression Analyses*

Regression analysis can be used to determine the power of an independent variable (X) to explain or predict a dependent variable (Y). We conducted regression analysis to discover the validity of our six hypotheses. From Section 5.3, the correlations between the four constructs were statistically significant. Therefore, in this section, we further investigated the relationships between the four variables, "Attention", "Relevance", "Confidence", and "Satisfaction". To verify the hypothesized relationships between the independent and dependent variables in our research framework, regression analysis was conducted between pairs of the aforementioned four factors.

### 5.4.1. Regression Results for **H1**

In particular, Table 7 presents the regression results used to test the hypotheses. A regression analysis using "Attention" as the independent variable and "Relevance" as the dependent variable showed that a larger standardized coefficient Beta assigned value indicated that the independent variable had a greater effect on the dependent variable and that attention had a highly significant effect on "Relevance", with a *p*-value of less than 0.001, signifying a significant level, and a value of 0.000 where the independent variable was predictive of the dependent variable at a marked level.

**Table 7.** Regression results for **H1**.

| | Standardized Coefficients β | Adjusted R Square | *F* | *t* | *p* |
|---|---|---|---|---|---|
| "Attention" | 0. 779 | 0.656 | 193.489 | 13.910 | 0.000 *** |

Variables of significance (*** $p \leq 0.001$).

### 5.4.2. Regression Results for **H2**, **H5**, and **H6**

The hypotheses **H2**, **H5,** and **H6** were tested by regressing "Attention", "Relevance", and "Confidence" on "Satisfaction". Table 8 provided results from the regression analysis for **H2**, **H5**, and **H6**. Regression analysis of the effect of "Attention", "Relevance", and "Confidence" on "Satisfaction" indicated that the standardized coefficients β of "Attention", "Relevance", and "Confidence" were 0.903 ($p < 0.001$), 0.920 ($p < 0.001$), and 0.830 ($p < 0.001$), respectively; "Attention", "Relevance", and "Confidence" have a significant influence on "Satisfaction". Thus, "Attention", "Relevance", and "Confidence" positively affected "Satisfaction".

**Table 8.** Regression results for **H2**, **H5**, and **H6**.

| | Standardized Coefficients β | Adjusted R Square | *F* | *t* | *p* |
|---|---|---|---|---|---|
| "Attention" | 0. 903 | 0.673 | 209.287 | 14.467 | 0.000 *** |
| "relevance" | 0.920 | 0.643 | *183.157* | *13.534* | 0.000 *** |
| "Confidence" | 0.830 | 0.607 | 157.796 | 12.522 | 0.000 *** |

Variables of significance (*** $p \leq 0.001$).

### 5.4.3. Regression Results for **H3** and **H4**

Table 9 provided results from the hypotheses **H3** and **H4** tested by regressing both "Attention" and "Relevance" on "Confidence". The results indicated that the standardized coefficient β of these two independent variables had a positive value, with an extremely strong *p*-value of less than 0.001, signifying a significant level. That is, the standardized coefficients β of "Attention" and "Relevance" were 0.758 ($p < 0.001$) and 0.833 ($p < 0.001$), respectively, both with a highly significant effect of "Attention" and "Relevance" on "Confidence".

**Table 9.** Regression results for **H3** and **H4**.

|  | Standardized Coefficients β | Adjusted *R* Square | *F* | *t* | *p* |
|---|---|---|---|---|---|
| "Attention" | 0. 758 | 0.534 | 116.754 | 10.805 | 0.000 *** |
| "relevance" | 0.833 | 0.594 | 148.936 | 12.204 | 0.000 *** |

Variables of significance (*** $p \leq 0.001$).

5.4.4. Hypotheses Results of the ARCS Model

In summary, quantitative analysis and a questionnaire were conducted on the basis of the ARCS model. The results as depicted in Table 10 indicated that students who gave a higher rating to "Attention" also had a more positive attitude toward "Relevance", "Confidence", and "Satisfaction". The stronger "Relevance" score among the students was correlated with more positive to "Confidence" and "Satisfaction". Furthermore, the results of hypothesis testing indicate that the ARCS model has predictive power, and this finding is consistent with the literature and is shown in Table 10.

**Table 10.** Hypotheses results of the ARCS model.

| Hypotheses | Results |
|---|---|
| **H1** | Supported *** |
| **H2** | Supported *** |
| **H3** | Supported *** |
| **H4** | Supported *** |
| **H5** | Supported *** |
| **H6** | Supported*** |

Significant at: *** $p \leq 0.001$.

## 6. Conclusions and Suggestions

One of the main contributions of this study is that we developed a cross-platform software system combining AR and chatbot, which is used as an auxiliary learning tool for junior high school biology courses. AR technology can instantly display 3D images and important features of a creature to help students easily understand the relevant information of the creature. Furthermore, this study uses the ARCS theoretical model proposed by Keller [54] as a research model to verify the effectiveness of integrating an AR-based chatbot into course design. Because the design of teaching materials is the main factor to attract students' attention and maintain interest in the learning process, if students have enough attention, they are interested in learning content or learning methods, and the learning effect will be good. Therefore, this study uses the Keller ARCS motivation model to explore the level of students' learning motivation responses in the four dimensions of introducing AR-based chatbot in biology courses, thereby increasing students' learning motivation. It is expected to create tools that are suitable for students to learn independently online. Finally, some conclusions and relevant suggestions are provided for future research.

### 6.1. Advantages of the AR-Based Chatbot System

- The chatbot designed for this study is quick to set up and easy to operate.
- The USDZ format, which can trigger AR, enriches the learning experience of students.
- Teaching webpage with guidelines on how to access the curriculum
- The online learning content of the natural science subject serves as a guide for higher education.
- The design of the course is structured and categorized in a systematic way to support different device experiences.

*6.2. ARCS Model Is Applicable in This Study*

In the digital learning environment, how to make learners focus on the learning content and maintain a long-lasting learning motivation is the difficulty that digital instructional designers must overcome when planning digital teaching materials. According to the definition of Keller's (1987) ARCS motivation theory, it is emphasized that the motivation of learners must be composed of these four elements to achieve the role of motivating learners to learn. After the experimental verification of this study, we not only proved that the AR-based chatbot provides a useful tool for middle school students to learn biology, but also after the experimental verification of this study, students generally affirmed the systematic teaching design of the AR-based chatbot in this study, which met the following criteria: (1) attention-grabbing, (2) teaching moderate material that connects with students' learning experience, so as to mobilize students' learning motivation, and (3) building confidence and a (4) sense of satisfaction. Keller [53] emphasized that the four factors, "Attention", "Relevance", "Confidence", and "Satisfaction", in the application of the motivation model are interlinked and that the positive orientation of each link will definitely make students' learning a virtuous circle.

The results of hypotheses **H1**, **H2,** and **H3** ("Attention" predicts "Relevance", "Confidence", and "Satisfaction") of this study show that there is a positive and significant relationship between students' attention motivation and related motivation in participating in courses. According to Keller's [54] motivation theory, attention to the motivational level is the primary condition for learning. Without attention, learners will not be able to arouse their subsequent learning motivation. This study also confirmed that attention motivation has a significant impact on related motivation. The course can improve students' learning motivation by combining interesting and attractive content with life-related issues. The maintenance of attention and the achievement of goals are beneficial to learning, and the course content related to the learners themselves can increase their interest in learning, and they can have confidence in themselves after learning and think that the success they have achieved through their own efforts is a pretty satisfying sense of accomplishment. Therefore, attention motivation to confidence motivation, attention motivation to satisfaction motivation, and related motivation to satisfaction motivation are all directly related to a series of learning process motivations in ARCS motivation. At the level of attention motivation, students arouse their own curiosity and attention because the course content is closely related to their own learning interests, as well as the course content and coursework needs. Secondly, on the level of relevant learning motivation, students said that, during the learning process, the course content can be connected with their past experience, and they will apply the knowledge they have learned in the classroom or in life, so that they will have a sense of personal relevance.

Hypotheses **H4** and **H5** ("Relevance" predicts "Confidence" and "Satisfaction") of this study also confirmed that students' participation in biology courses combined with the AR-based chatbot system has a significant impact on confidence. The course can discuss topics closely related to biology and life, share successful experiences, and make students feel relevant and willing to participate, thereby enhancing students' learning motivation. Through the biology courses combined with AR-based chatbot system, rich and interesting course content, such as 3D animation teaching, is combined in the course to attract students' attention. It creates a sense of relevance, in line with personal interests and expectations, believing that one can obtain the expected learning results through hard work in the course, obtain a high score to build course-learning confidence, and finally obtain internal or external encouragement due to achievements, resulting in a sense of satisfaction and a desire to continue learning.

Furthermore, **H6** ("Confidence" predicts "Satisfaction"): The results of the research hypothesis show that there is a positive and significant relationship between the confidence motivation and satisfaction motivation of students participating in the course. When the obtained learning achievement is consistent with the expectation, the learner's learning motivation can be further increased. This study also confirmed that confidence motivation

has a significant impact on satisfaction motivation when students participate in biology courses combined with the AR-based chatbot system. The course can make it easy for students to learn and understand the knowledge and content acquired in the course through a relaxed learning method. They are successfully used in the classroom time, that is, to meet the needs of learners, and finally to improve students' learning motivation. Furthermore, at the level of confidence and learning motivation, students feel that the difficulty of the course content is within the scope of their own control and that they can apply the knowledge they have learned, making them believe that they can master the key points of the course with their own efforts and obtain the expected scores, so as to build confidence in learning. Finally, on the level of satisfying learning motivation, students feel satisfied with the knowledge they have acquired in the course, being able to achieve their personal learning goals and receiving the encouragement and feedback from teachers to generate a sense of learning accomplishment.

### 6.3. Some Suggestions

A software improvement survey through the ARCS motivational model revealed that the students' learning experience not only generated interest but also fostered learning attitudes. Student system improvement surveys exhibited a high degree of positive correlation, with significant levels of data for each tested hypothesis in the regression analysis, wherein the dependent variables were predicted by the independent variables. The gender of students using this type of e-learning tool may lead to a difference in their interest in learning. The survey also found that an in-depth study of whether the length of time spent on pre-learning the natural science subject or online learning affects the effectiveness of student learning would be a key to system improvement.

To provide students with an initial understanding of how AR and chatbots work, the educational platform for chatbots uses only Facebook Messenger as an operating system. It is hoped that more advanced educational platforms will be developed in the future to include familiar software systems, such as Line and Instagram, to support the use of cross-platform devices, so that users can operate on more diverse platforms. The software should include a time management function for learning in the future, or the total learning time for each chapter should be flexibly adjusted in the design of the teaching materials, so that the expected effect will be different. When learning biology with AR digital textbooks, they can use their smart device for interaction and reinforce the subject content on the screen to deepen the connection to knowledge. Through interesting extracurricular videos, students are brought closer to the subjects they are studying, with the aim of providing a new type of online learning tool in the pandemic era, using chatbots as a different mode of learning at home.

This study and the administration of the questionnaires were conducted before the outbreak of the local infection cases in Taiwan. Due to the constraints of location and target respondents, schools at all levels nationwide were closed, and most students studied online at home. As a result, this study only involved students in the first year of junior high school and did not extend to students in the second and third years. The operational data on the age of the respondents cannot be fully obtained, and the final results may differ from the prior research design. It would have been useful to include parents and teachers in the study to observe and conduct a survey on teaching opinions. It is recommended that, when the outbreak subsides later, students return to school to fill out the questionnaire again, as they will have had more time to fully experience the system during their online learning at home, and the results may be different, or the study participants be included in the control group of those who are using the platform or not, so that the context of the students' online learning with the system can be better understood. The above suggestions are offered as a reference for future research and development of software.

**Author Contributions:** Conceptualization, C.-H.C., Y.-K.W. and J.-H.L.; methodology, C.-H.C., Y.-K.W. and J.-H.L.; software, Y.-K.W.; validation, C.-H.C. and J.-H.L.; investigation, Y.-K.W.; data curation, Y.-K.W.; writing—original draft preparation, Y.-K.W. and J.-H.L.; writing—review and editing, C.-H.C. and J.-H.L.; visualization, C.-H.C. and J.-H.L.; supervision, J.-H.L.; project administration, J.-H.L.; funding acquisition, J.-H.L. All authors have read and agreed to the published version of the manuscript.

**Funding:** This research was funded by National Science and Technology Council of Taiwan, under grant no. [MOST 110-2221-E-431-001], and [MOST 111-2221-E-431-001].

**Data Availability Statement:** The data presented in this study are available on request from the corresponding author.

**Acknowledgments:** This research was supported by NSTC of Taiwan, under grant no. [MOST 110-2221-E-431-001] and [MOST 111-2221-E-431-001].

**Conflicts of Interest:** The authors declare no conflict of interest.

## Appendix A

**Table A1.** Profile of participants.

| Dimension | Question | | Frequency | Percentage |
|---|---|---|---|---|
| Personal information | Gender | Male | 48 | 47.1 |
| | | Female | 54 | 52.9 |
| | How much time do you spend on studying biology lessons per day? | 0.5~1 h | 65 | 63.7 |
| | | 1~1.5 h | 26 | 25.4 |
| | | 1.5 h or more | 11 | 10.7 |
| | Have you ever used chatbots from communication software before? | Yes | 54 | 52.9 |
| | | No | 48 | 47.0 |
| | How much time do you typically spend using chatbots? | 0.5~1 h | 36 | 66.7 |
| | | 1~1.5 h | 11 | 20.3 |
| | | 1.5 h or more | 7 | 13.0 |
| Learning ability | I find it easy to study biology classes | Strongly agree | 14 | 13.7 |
| | | Agree | 19 | 18.6 |
| | | Neither agree nor disagree | 52 | 51.0 |
| | | Disagree | 15 | 14.7 |
| | | Strongly disagree | 2 | 2.0 |
| | As long as I work hard, I can learn biology subjects well. | Strongly agree | 32 | 31.4 |
| | | Agree | 49 | 48 |
| | | Neither agree nor disagree | 17 | 16.7 |
| | | Disagree | 4 | 3.9 |
| | | Strongly disagree | 0 | 0 |
| | I usually do well in biology classes. | Strongly agree | 12 | 11.8 |
| | | Agree | 23 | 22.5 |
| | | Neither agree nor disagree | 43 | 42.2 |
| | | Disagree | 21 | 20.6 |
| | | Strongly disagree | 3 | 2.9 |

**Table A1.** *Cont.*

| Dimension | Question | | Frequency | Percentage |
|---|---|---|---|---|
| Learning ability | I believe in my ability to teach my classmates and solve problems. | Strongly agree | 12 | 11.8 |
| | | Agree | 24 | 23.5 |
| | | Neither agree nor disagree | 36 | 35.3 |
| | | Disagree | 28 | 27.5 |
| | | Strongly disagree | 2 | 2.0 |
| | I think I study biology classes better than my classmates. | Strongly agree | 7 | 6.9 |
| | | Agree | 20 | 19.6 |
| | | Neither agree nor disagree | 42 | 41.2 |
| | | Disagree | 24 | 23.5 |
| | | Strongly disagree | 9 | 8.8 |
| Application ability | It is helpful for me to study biology subjects | Strongly agree | 30 | 29.4 |
| | | Agree | 42 | 41.2 |
| | | Neither agree nor disagree | 27 | 26.5 |
| | | Disagree | 3 | 2.9 |
| | | Strongly disagree | 0 | 0 |
| | I use biology in my life. | Strongly agree | 27 | 26.5 |
| | | Agree | 39 | 38.2 |
| | | Neither agree nor disagree | 33 | 32.4 |
| | | Disagree | 3 | 2.9 |
| | | Strongly disagree | 0 | 0 |
| | Studying biology allows me to solve problems of everyday life. | Strongly agree | 17 | 17.6 |
| | | Agree | 44 | 43.1 |
| | | Neither agree nor disagree | 31 | 30.4 |
| | | Disagree | 9 | 8.8 |
| | | Strongly disagree | 0 | 0 |
| | Studying biology is very helpful for me in high school in the future | Strongly agree | 27 | 26.5 |
| | | Agree | 47 | 46.1 |
| | | Neither agree nor disagree | 26 | 25.5 |
| | | Disagree | 2 | 2 |
| | | Strongly disagree | 0 | 0 |
| | Studying biology contributes to the study of other subjects | Strongly agree | 17 | 16.7 |
| | | Agree | 31 | 30.4 |
| | | Neither agree nor disagree | 47 | 46.1 |
| | | Disagree | 7 | 6.9 |
| | | Strongly disagree | 0 | 0 |
| Problem-solving ability | I can learn many interesting things from biology. | Strongly agree | 40 | 39.2 |
| | | Agree | 40 | 39.2 |
| | | Neither agree nor disagree | 22 | 21.6 |
| | | Disagree | 0 | 0 |
| | | Strongly disagree | 0 | 0 |

**Table A1.** *Cont.*

| Dimension | Question | | Frequency | Percentage |
|---|---|---|---|---|
| Problem-solving ability | I will actively look for opportunities to learn biology to improve my knowledge | Strongly agree | 9 | 8.8 |
| | | Agree | 28 | 27.5 |
| | | Neither agree nor disagree | 49 | 48 |
| | | Disagree | 14 | 13.7 |
| | | Strongly disagree | 2 | 2 |
| | When I encounter a biology subject problem that I am not sure about, I will use various resources. | Strongly agree | 21 | 20.6 |
| | | Agree | 39 | 38.2 |
| | | Neither agree nor disagree | 37 | 36.3 |
| | | Disagree | 5 | 4.9 |
| | | Strongly disagree | 0 | 0 |
| | In addition to what is taught in school, I will take the initiative to explore things and use the knowledge I have learned in biology class to solve problems. | Strongly agree | 16 | 15.7 |
| | | Agree | 33 | 32.4 |
| | | Neither agree nor disagree | 43 | 42.2 |
| | | Disagree | 8 | 7.8 |
| | | Strongly disagree | 2 | 2 |
| | When encountering biology that I don't know, I will take the initiative to ask the teacher | Strongly agree | 19 | 18.6 |
| | | Agree | 34 | 33.3 |
| | | Neither agree nor disagree | 44 | 43.1 |
| | | Disagree | 5 | 4.9 |
| | | Strongly disagree | 0 | 0 |

**Table A2.** Participants' answers of the ARCS model.

| Dimension | Question | | Frequency | Percentage |
|---|---|---|---|---|
| "Attention" | Learning in a chatbot way gets my attention. | Strongly agree | 34 | 33.3 |
| | | Agree | 41 | 40.2 |
| | | Neither agree nor disagree | 26 | 25.5 |
| | | Disagree | 1 | 1.0 |
| | | Strongly disagree | 0 | 0 |
| | When using chatbots to learn about biology, it helps me to focus and I am less prone to distraction. | Strongly agree | 28 | 27.5 |
| | | Agree | 35 | 34.3 |
| | | Neither agree nor disagree | 33 | 32.4 |
| | | Disagree | 5 | 4.9 |
| | | Strongly disagree | 1 | 1.0 |
| | Chatbots with biology themes are fun. | Strongly agree | 35 | 34.3 |
| | | Agree | 45 | 44.1 |
| | | Neither agree nor disagree | 19 | 18.6 |
| | | Disagree | 3 | 2.9 |
| | | Strongly disagree | 0 | 0 |

<p align="center">**Table A2.** *Cont.*</p>

| Dimension | Question | | Frequency | Percentage |
|---|---|---|---|---|
| "Attention" | I can concentrate better while studying biology with a chatbot than with a book. | Strongly agree | 33 | 32.4 |
| | | Agree | 35 | 34.3 |
| | | Neither agree nor disagree | 28 | 27.5 |
| | | Disagree | 4 | 3.9 |
| | | Strongly disagree | 2 | 2.0 |
| | For the class of biological classification, I will be interested in watching extra-curricular videos. | Strongly agree | 39 | 38.2 |
| | | Agree | 34 | 33.3 |
| | | Neither agree nor disagree | 25 | 24.5 |
| | | Disagree | 4 | 3.9 |
| | | Strongly disagree | 0 | 0 |
| "Relevance" | In the course of biology-classification, using AR to manipulate animals allows me to better understand its appearance. | Strongly agree | 49 | 48.0 |
| | | Agree | 37 | 36.3 |
| | | Neither agree nor disagree | 16 | 15.7 |
| | | Disagree | 0 | 0 |
| | | Strongly disagree | 0 | 0 |
| | Digital content is so interesting to me, I hope there are more digital teaching materials and use chatbots to learn. | Strongly agree | 35 | 34.3 |
| | | Agree | 36 | 35.3 |
| | | Neither agree nor disagree | 27 | 26.5 |
| | | Disagree | 4 | 3.9 |
| | | Strongly disagree | 0 | 0 |
| | For the class of biology-classification, for me, extra-curricular videos can better understand the biological activities. | Strongly agree | 34 | 33.3 |
| | | Agree | 39 | 38.2 |
| | | Neither agree nor disagree | 27 | 26.5 |
| | | Disagree | 2 | 2.0 |
| | | Strongly disagree | 0 | 0 |
| | Using chatbots to learn about biology has given me a new way of learning. | Strongly agree | 38 | 37.3 |
| | | Agree | 37 | 36.3 |
| | | Neither agree nor disagree | 24 | 23.5 |
| | | Disagree | 2 | 2.0 |
| | | Strongly disagree | 1 | 1.0 |
| | Using chatbots to learn natural course content is something I haven't tried in the past. | Strongly agree | 47 | 46.1 |
| | | Agree | 29 | 28.4 |
| | | Neither agree nor disagree | 18 | 17.6 |
| | | Disagree | 6 | 5.9 |
| | | Strongly disagree | 2 | 2.0 |
| "Confidence" | I have the confidence to teach students who don't know how to use the interface of chatbots. | Strongly agree | 18 | 17.6 |
| | | Agree | 33 | 32.4 |
| | | Neither agree nor disagree | 36 | 35.3 |
| | | Disagree | 12 | 11.8 |
| | | Strongly disagree | 3 | 2.9 |

**Table A2.** *Cont.*

| Dimension | Question | | Frequency | Percentage |
|---|---|---|---|---|
| "Confidence" | In the process of operating a chatbot, it is easy to use, not by luck. | Strongly agree | 26 | 25.5 |
| | | Agree | 37 | 36.3 |
| | | Neither agree nor disagree | 31 | 30.4 |
| | | Disagree | 7 | 6.9 |
| | | Strongly disagree | 1 | 1.0 |
| | I like the way of online digital learning using this chatbot to learn the lessons well. | Strongly agree | 27 | 26.5 |
| | | Agree | 38 | 37.3 |
| | | Neither agree nor disagree | 31 | 30.4 |
| | | Disagree | 4 | 3.9 |
| | | Strongly disagree | 2 | 2.0 |
| | I actively learn to explore the interface buttons to help interact with the chatbot. | Strongly agree | 25 | 24.5 |
| | | Agree | 39 | 38.2 |
| | | Neither agree nor disagree | 32 | 31.4 |
| | | Disagree | 6 | 5.9 |
| | | Strongly disagree | 0 | 0 |
| | I take a class on a chatbot and I can understand what it means. | Strongly agree | 24 | 23.5 |
| | | Agree | 36 | 35.3 |
| | | Neither agree nor disagree | 38 | 37.3 |
| | | Disagree | 4 | 3.9 |
| | | Strongly disagree | 0 | 0 |
| "Satisfaction" | After using the digital teaching materials of chatbots, learning makes me understand biology better and increase my sense of accomplishment. | Strongly agree | 32 | 31.4 |
| | | Agree | 34 | 33.3 |
| | | Neither agree nor disagree | 30 | 29.4 |
| | | Disagree | 6 | 5.9 |
| | | Strongly disagree | 0 | 0 |
| | For the biology-classification course, I use AR to manipulate animals, and I am more patient to invest time. | Strongly agree | 34 | 33.3 |
| | | Agree | 36 | 35.3 |
| | | Neither agree nor disagree | 28 | 27.5 |
| | | Disagree | 4 | 3.9 |
| | | Strongly disagree | 0 | 0 |
| | Build a chatbot on Messenger, which I will use as a review. | Strongly agree | 33 | 32.4 |
| | | Agree | 27 | 26.5 |
| | | Neither agree nor disagree | 36 | 35.3 |
| | | Disagree | 5 | 4.9 |
| | | Strongly disagree | 1 | 1.0 |
| | I am happy using chatbots for digital learning. | Strongly agree | 32 | 31.4 |
| | | Agree | 34 | 33.3 |
| | | Neither agree nor disagree | 31 | 30.4 |
| | | Disagree | 5 | 4.9 |
| | | Strongly disagree | 0 | 0 |

**Table A2.** *Cont.*

| Dimension | Question | | Frequency | Percentage |
|---|---|---|---|---|
| "Satisfaction" | When encountering biology that I don't know, I will take the initiative to ask the teacher | Strongly agree | 39 | 38.2 |
| | | Agree | 33 | 32.4 |
| | | Neither agree nor disagree | 26 | 25.5 |
| | | Disagree | 4 | 3.9 |
| | | Strongly disagree | 0 | 0 |

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
