# Peer review of "Integrating Chatbot and Augmented Reality Technology into Biology Learning during COVID-19"

_electronics, doi:10.3390/electronics12010222_

Round 1
Reviewer 1 Report
In Table 9, why adjusted R2 is used instead of R2? RMSE metrics can also be used.
Why pearson correlation results is used? Kendall and Spearman correlation also availabe?
What are the limitations of the study?
Include performance of the system in terms of computation time
Reviewer 2 Report
The study presents a cross-platform software system combining AR and chatbot, which is used as a learning aid tool for high school biology courses. This study uses the ARCS theoretical model proposed by Keller as a research model to verify the effectiveness of AR-based chatbot integration in course design.
The topic presented is relevant and well written.
It is possible to follow the communication flow easily The literature review is adjusted to the topic and thendiscussion debates the results in view of the literature presented.The conclusions are succinct and correctly close the text.
Some suggestions to improve the paper:
- Check the hyphenation of words throughout the text;
- Define the acronym VR in line 66;
- Define the acronym ARCS on line 116;
- In line 125 the word "figure" appears in small letters, but in the rest of the text "Figure" appears;
- In line 243 the initial word ("rule") must start with a capital letter;
- Define the API acronym on line 253;
- Change the sentence of lines 299-302, since it has the word "so" is repeated;
- Missing a final point on line 388 and line 591;
- Replace the 2 dots with a semicolon on line 561;
- In line 673 and 679, the authors must want to refer to Figure 5 and not to Figure 4;
-Table 8 is broken into two different pages. Place the table on only one page;
- On line 814 remove the extra "s" in the sentence "...their own efforts s is a pretty satisfying...";
- On line 845 add a blank space after H5;
- In line 834 change the "A" to "a" in the sentence "...to achievements, resulting in A sense of satisfaction...";
Reviewer 3 Report
General comments:
The current research is an extremely interesting one in the context of recent global events, which undoubtedly affected all spheres of action of the human being, including teaching and learning processes.
By combining theoretical and empirical research, the authors managed to bring to the attention of the audience some notable aspects regarding alternative learning methods, in a situation characterized by uncertainty, focusing on the analysis of one specific educational tool.
To a large extent, the research is correctly structured, following a natural flow and presenting key aspects undoubtedly targeted by the audience. Moreover, the results obtained arouse the attention and interest of the audience. At the same time, the usage of tables and figures is useful in presenting the analysis' main findings.
Awareness of the limitations of research is a strong point of it, as it lays the foundations for any possible future work.
Recommendations:
- It is recommended to check the indentation on the left for the numbered list between lines 114-117;
- In the current format of the paper, a probably unintentional error could be observed, related to the title of table 2 (Table 2. KMO and Bartlett's sphericity test results), which is positioned on a separate page from the table itself;
- Despite the fact that it is mentioned in the text, Figure 4 was not found in the content of the manuscript. Therefore, it is recommended to check the numbering of the figures;
- There is also the possibility that the numbering of the tables or their citation in the text to be incorrect. Lines 549-550 mention: The steps we developed are described in more detail in Table 2. According to the content of the manuscript, Table 1 seems more appropriate for the description.
- Minor discrepancies were observed between the present paper and required template of the journal. Therefore, it would be advisable for the author/s to review the structure and the design of the specific parts within the manuscript.
Reviewer 4 Report
As we continue to explore technologies for learning, the topic addressed in the paper is of interest. The framework used for the research is suitable and the analysis of data completed well. However, to enrich the study, or similar studies, consider the use of mixed methods. Some interviews with students can provide greater depth of insight into the effect of the tools. In the study limitations also consider longitudinal studies for future research. The effect of novelty should be considered. Also, for further research, consider other methods to assess the learning outcomes of students. It's important to highlight the constraints of the study.
Overall the study is interesting and quite well structured. The author/s have avoided over-generalizations, which is good.
On the technical aspect of writing: the writing is quite good overall. However, it can be difficult to read. One major area to improve is on sentence construction. There are many long sentences. These sentences make it difficult to digest meaning. Paragraphing can also be improved. There are some long paragraphs. As a guide, one paragraph = one key idea.
